# Study of a SiC Trench MOSFET Edge-Termination Structure with a Bottom Protection Well for a High Breakdown Voltage

**Jee-Hun Jeong, Ju-Hong Cha, Goon-Ho Kim, Sung-Hwan Cho and Ho-Jun Lee \***

Department of Electrical Engineering, Pusan National University, Pusan 46241, Korea;
hemn4221@gmail.com (J.-H.J.); ciwsssh@nate.com (J.-H.C.); windthink@pusan.ac.kr (G.-H.K.);
tjdghks1027s@gmail.com (S.-H.C.)
**\*** Correspondence: hedo@pusan.ac.kr; Tel.: +82-51-510-2301

**Abstract:** A novel edge-termination structure for a SiC trench metal–oxide semiconductor field-effect transistor (MOSFET) power device is proposed. The key feature of the proposed structure is a periodically formed SiC trench with a bottom protection well (BPW) implantation region. The trench can be filled with oxide or gate materials. Indeed, it has almost the same cross-sectional structure as the active region of a SiC trench MOSFET. Therefore, there is little or no additional process loads. A conventional floating field ring (FFR) structure utilizes the spreading of the electric field in the periodically depleted surface region formed between a heavily doped equipotential region. On the other hand, in the trenched ring structure, an additional quasi-equipotential region is provided by the BPW region, which enables deeper and wider field-spreading profiles, and less field crowding at the edge region. The two-dimensional Technology Computer Aided Design (2D-TCAD) simulation results show that the proposed trenched ring-edge termination structures have an improved breakdown voltage compared to the conventional floating field ring structure.

**Keywords:** breakdown voltage; edge termination; conventional floating field ring; SiC trench structure; bottom protection well

---

## 1. Introduction

Many reports on the development of wide band gap materials and the design of novel edge-termination structures for a high breakdown voltage have been published. Several structures have been investigated to spread the electric field at the edge of the corner [1]. One example is a conventional floating field ring (FFR) structure with a periodically heavily doped region, which can be made in the same processing step as the active region during device fabrication. This structure is used to improve the breakdown voltage characteristics by shifting the maximum electric field from the surface to the substrate and reducing the peak value [1–3]. On the other hand, when a high voltage structure is designed, this edge structure is needed for the many rings due to spreading of the electric field.

This study examined the design of novel edge-termination structures, including the trenched ring structure filled with polysilicon or oxide. The poly-trenched structure was formed using the same process as that for the active region. The newly proposed structures contained heavy doping at the bottom of the trench to reduce the crowding electric field [3–10] and to add a quasi-equipotential region. By varying the trench width and ring space, these structures enable deeper and wider field-spreading profiles at the edge region and improve the breakdown voltage compared to the FFR structure as the same cross-section.

## 2. Device Structure and Models

The FFR and trench edge-termination structures were simulated using a Silvaco-TCAD two dimensional (2D) simulator. In these structures, with a total length of 30 μm, the $S$ and $W$ parameters are the ring length and the distance between a ring and the next ring (or trench width at a trenched structure), respectively. The thickness and doping concentration of the drift region are $t_{epi}$ and $n_{epi}$, respectively, and $n_{ring}$, $n_{BPW}$ are the doping concentrations of the ring and bottom protection well (BPW), respectively, as indicated in Table 1. These parameters are considered by using a depletion width equation.

$$W = \sqrt{\frac{2\varepsilon_s V_{bi}}{q}\left(\frac{n_{ring} + n_{epi}}{n_{ring} n_{epi}}\right)} \tag{1}$$

**Table 1.** Geometry parameters and conditions for the edge-termination structures used in the simulations.

| Geometry Parameter | Value |
|---|---|
| Thickness and doping concentration of drift region, $t_{epi}/n_{epi}$ | 15 μm/$4 \times 10^{15}$ cm$^{-3}$ |
| Doping concentration of p+ ring, $n_{ring}$ | $1 \times 10^{20}$ cm$^{-3}$ |
| Doping concentration of bottom protection well, $n_{BPW}$ | $1 \times 10^{18}$ cm$^{-3}$ |
| Ring length, $S$ | 0.5~1.5 μm (±0.2 μm) |
| A distance between rings, $W$ | 0.5~1.5 μm (±0.2 μm) |
| Total $S + W$ length | 2.5 μm |

The depletion length is calculated about 0.9 μm between the drift region and the ring region under a non-zero bias condition. The space and width parameters are varied to find a punch-through condition.

This paper describes four types of edge-termination structures, as shown in Figure 1. The conventional FFR structure, Figure 1a, formed the periodically heavily doped rings at the surface. The trenched structures are filled with polysilicon or oxide implant Al at the bottom of the trench, as indicated in Figure 1b,c, respectively. Al implantation for BPW formation was simulated using Monte Carlo models by Silvaco-TCAD. This edge-termination combines the FFR with the trenched structure (Figure 1d). A conventional FFR five-ring structure and proposed trenched five-ring structures were designed.

All edge-termination structures mentioned above were simulated by varying the $S$ and $W$ parameters. Numerical simulations were performed to analyze the breakdown voltage of these structures. This simulation of a reverse bias was considered by SiC ionization coefficients, which were used by Chynoweth [11].

$$\alpha_{n,p} = a_{n,p} \exp\left[-\frac{b_{n,p}}{E}\right] \tag{2}$$

where $E$ represents the value of the electric field, $a_{n,p}$ are the electron and hole ionization fitting coefficient, and $b_{n,p}$ are the electron and hole critical electric field. The magnitudes of the electron ionization parameters $a_n$ and $b_n$ used in these simulations are $1.66 \times 10^6$ cm$^{-1}$ and $6.6 \times 10^6$ Vcm$^{-1}$, respectively, and the values of the hole ionization parameters $a_p$ and $b_p$ used in these simulations are $5.18 \times 10^6$ cm$^{-1}$ and $1.47 \times 10^6$ Vcm$^{-1}$, respectively.

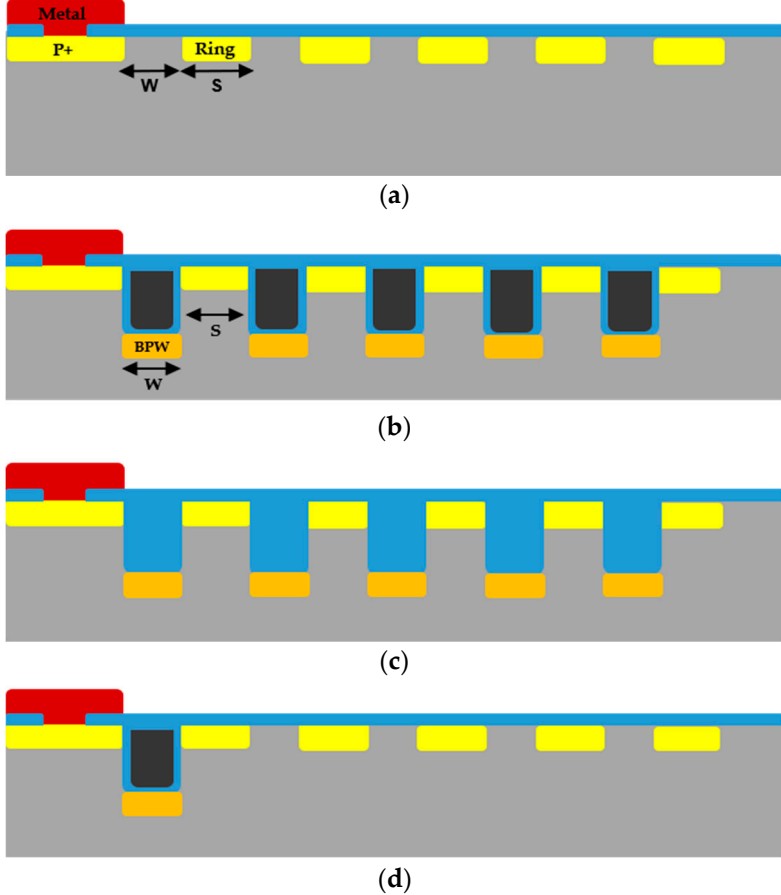

**Figure 1.** (**a**) Conventional floating field ring (FFR) edge-termination with a five-ring structure, (**b**) and (**c**) The poly-trench and oxide trench structure filled with polysilicon and oxide, respectively, and (**d**) The edge termination that mixes the FFR and trenched structure.

The breakdown voltage for a cylindrical junction can be calculated by solving Poisson's equation in cylindrical coordinates [1,3].

$$\frac{1}{r}\frac{d}{dr}\left(r\frac{dV}{dr}\right) = -\frac{1}{r}\frac{d}{dr}(rE) = -\frac{Q(r)}{\varepsilon_s} \tag{3}$$

where the potential, $V$, and electric field, $E$, are a function of the junction radius. The electric distribution equation can be obtained by integrating the above equation with the boundary equation in that the electric field should be zero when the junction radius is equal to the depletion length ($r_d$):

$$E(r) = \frac{qN_d}{2\varepsilon_s}\left(\frac{r^2 - r_d^2}{r}\right). \tag{4}$$

The breakdown voltage for the cylindrical junction can be obtained by integrating the above equation for the electric field distribution:

$$V(r) = \frac{qN_d}{2\varepsilon_s}\left[\left(\frac{r_J^2 - r^2}{2}\right) + r_d^2\ln\left(\frac{r}{r_J}\right)\right]. \tag{5}$$

The breakdown voltage for the cylindrical junction is a function of the radius of curvature. In the case of a power device, it requires a large junction depth to reduce the degradation of the breakdown voltage [1,3].

In this study, an additional trenched structure raised the junction radius and formed a quasi-equipotential region at the bottom of the trench. Therefore, the breakdown voltage for the cylindrical junction increases with increasing radius of the junction curvature [1,12].

## 3. Simulation Results and Discussion

### 3.1. Breakdown Voltage Characteristics and Potential Distributions

Breakdown voltages for the conventional FFR and proposed trenched edge-termination structures are shown in Figure 2. It has been reported that the breakdown voltage is affected by an expansion ratio of the space or width and a slightly increasing expansion ratio is beneficial for a higher breakdown voltage [3]. If the problem includes a structural variation, the space/width ratio, and the expansion ratio, it becomes too complex. In this study, the simulation was done with a fixed periodic length of the ring structure, which enables a clearer analysis of the effects of termination structures.

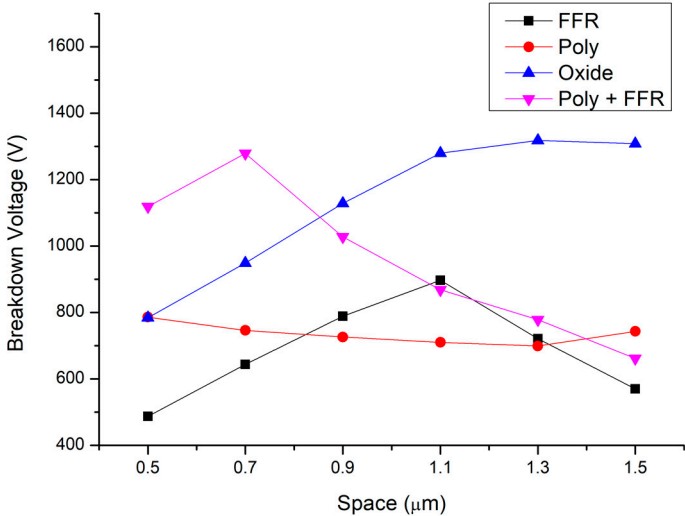

**Figure 2.** Breakdown voltages of various edge-termination structures versus different spaces.

Most breakdown voltages of trenched structures are higher than that of the conventional FFR structure. This means that the trenched structure is driving the potential distribution more efficiently because the junction curvature is increased.

As expected, for the conventional FFR structure shown in Figure 2, the breakdown voltage increases and then decreases again with increasing space (decreasing width). For a shorter space region, the punch-through does not occur, or the effect of the punch-through on the enhancing curvature of the equipotential line is insufficient to obtain an optimal condition. On the other hand, for a longer space region, the breakdown voltage decreases due to the increase in the equipotential region provided by the P+ doped ring structure. The effects of enlarging the radius of the junction curvature, by introducing a combined BPW and ring-doping structure, are clearly shown in Figure 3d. Because the depletion region can be formed between the BPW and the first P+ ring, the highest breakdown voltage is already obtained with the short space. For the polysilicon trench structure, the breakdown voltage is almost independent of the space/width variation. We can easily expect this trend because this structure provides an equipotential state to the entire trench region. There is nearly no potential difference along the ring structure. The reverse bias voltage is sustained by the last BPW and the P+ ring alone, as shown in Figure 3b. The breakdown voltage of the combined the FFR and trenched structure is quite higher than that of the FFR or the poly trench structure as shown in Figure 2. Because this structure combines the effect of the FFR and the poly-trench. It should be noted that no additional process step is necessary for the poly-trench with the FFR. The oxide trench shows the highest breakdown

voltage within our simulation cases. The results probably come from the combined effect of the voltage difference within the oxide region and the increasing junction curvature.

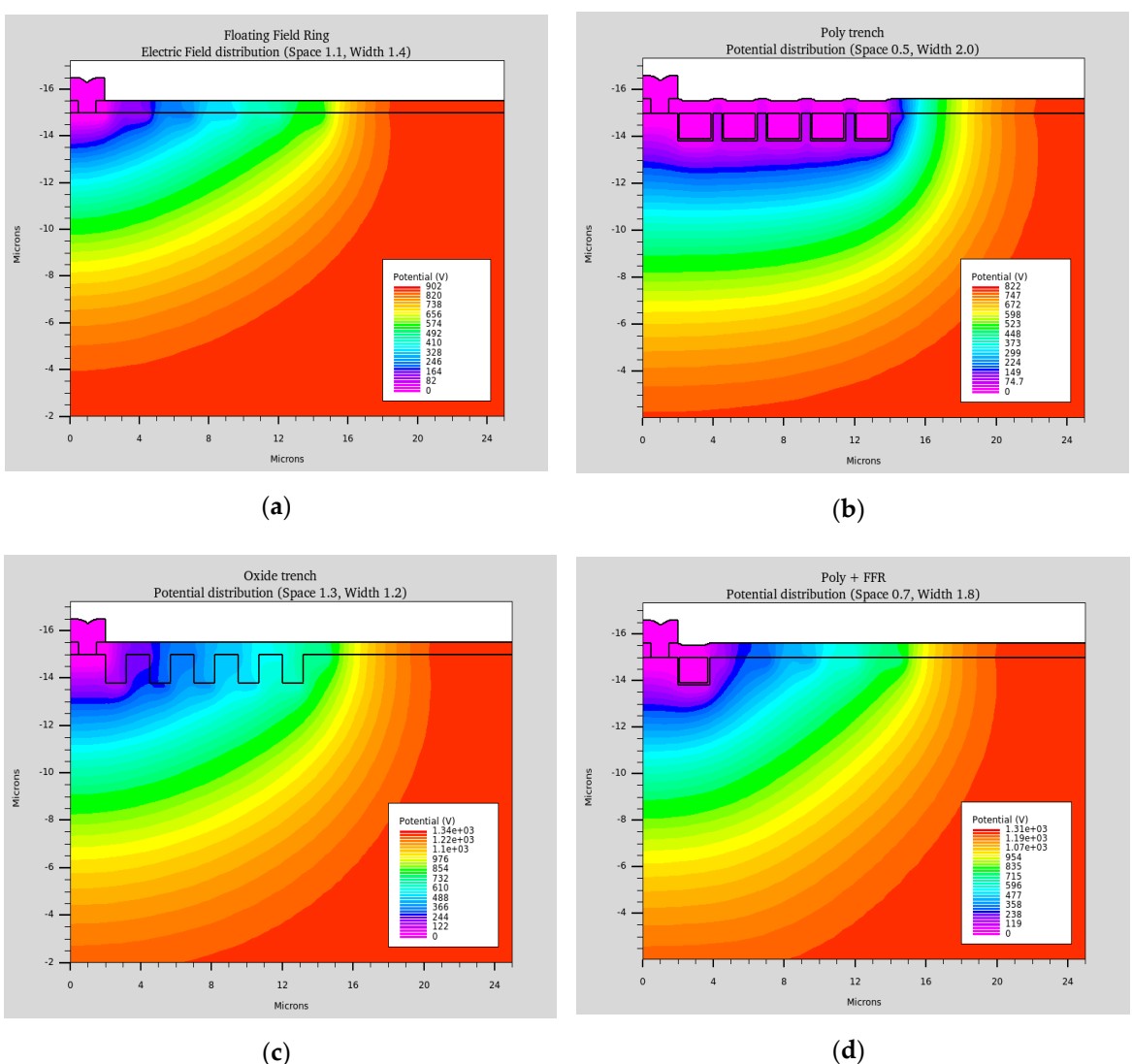

**Figure 3.** (**a**) Potential distributions close to the breakdown voltage of the conventional FFR (space, 1.1 μm; width, 1.4 μm), (**b**) The poly-trench (space, 0.5 μm; width, 2.0 μm), (**c**) The oxide trench (space, 1.3 μm; width, 1.2 μm), and (**d**) A combination of the poly-trench and the FFR (space, 0.7 μm; width, 1.8 μm).

### 3.2. Electric Field Distribution of Edge-Termination Structures

Figure 4 shows the three-dimensional electric field distributions of the four edge-termination structures under avalanche breakdown conditions.

Figure 4a presents the electric field distribution of the conventional FFR. This structure that optimized the space and width parameters supports the highest electric field ($E_{max} = 3.95\,\text{MV/cm}$) by five rings. On the other hand, the poly-trench supports the highest electric field ($E_{max} = 2.97\,\text{MV/cm}$). Although the maximum electric field of the poly-trench is lower than that of the conventional FFR, the breakdown voltage of the conventional FFR is much higher. This is because, in the poly-trench, only the last trench supports the electric field, as shown in Figure 4b. The oxide trench and combined poly-trench with FFR structures maintain the highest electric field: $E_{max} = 3.3\,\text{MV/cm}$ and $E_{max} = 3.47\,\text{MV/cm}$, respectively. Therefore, all junctions can support the electric field by five

trenches, as shown in Figure 4c,d. Thus, the breakdown voltage characteristics of both structures are higher than those of the conventional FFR.

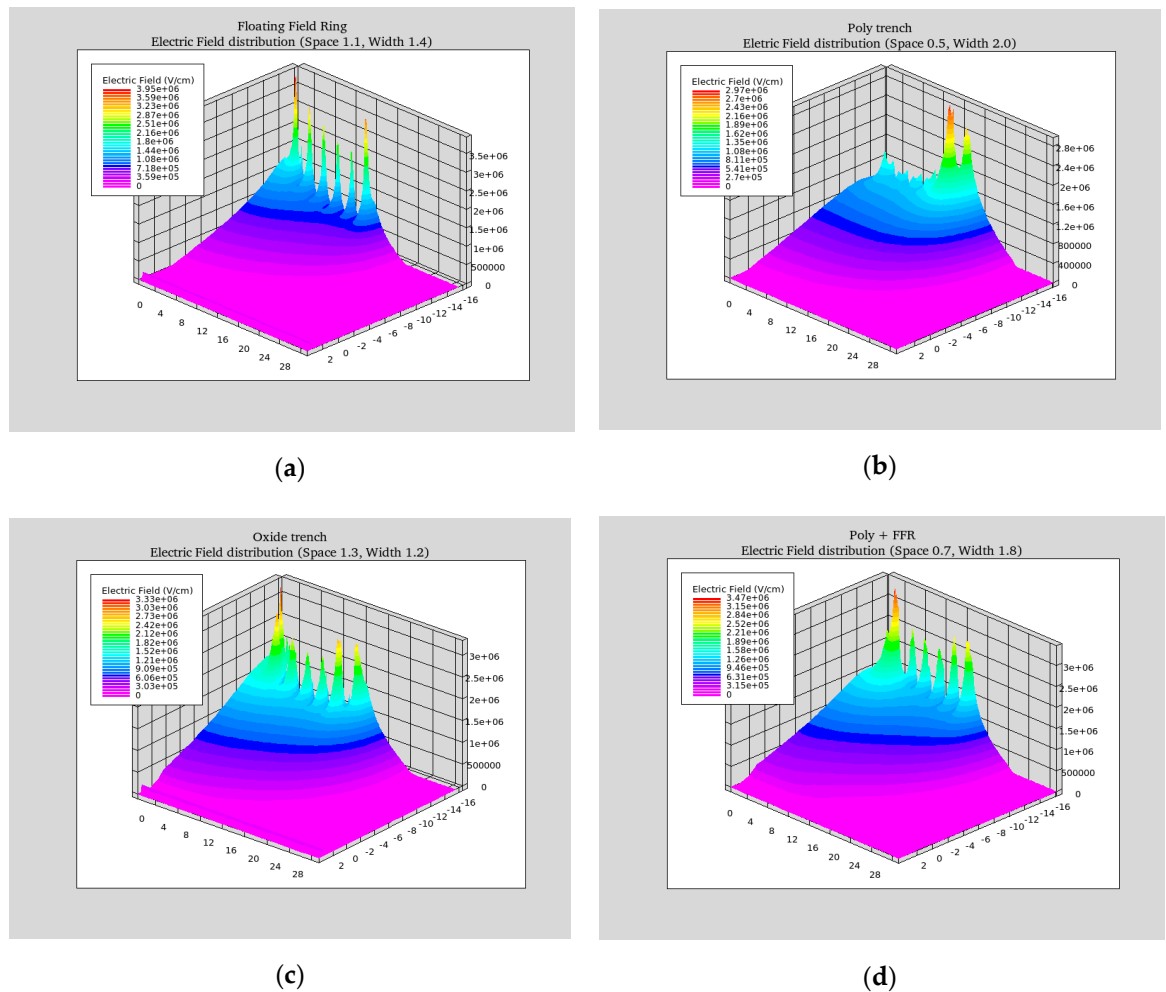

(**a**)  (**b**)

(**c**)  (**d**)

**Figure 4.** (**a**) Electric field distribution of the conventional FFR ($E_{max}$ = 3.95 MV/cm), (**b**) Electric field distribution of the trenched structure filled with polysilicon ($E_{max}$ = 2.97 MV/cm), (**c**) Electric field distribution of the trenched structure filled with oxide ($E_{max}$ = 3.33 MV/cm), and (**d**) Electric field distribution of a combination of the poly-trench and the conventional FFR ($E_{max}$ = 3.47 MV/cm).

## 4. Conclusions

Novel edge-termination structures were proposed to improve the breakdown voltage characteristics. The proposed structures consist of a polysilicon trench, an oxide trench, and a mixed poly-trench with an FFR structure. They were formed by implanting Al at the bottom of the trench. The effects of this structure increase the cylindrical junction curvature and enable a deeper and wider potential profile.

Numerical simulations were performed to analyze the breakdown voltage of the FFR and the proposed trenched structures by varying the space and width parameters. The simulation results showed that the highest breakdown voltage of the FFR structure was 990 V (with a space and width of 1.1 μm and 1.4 μm, respectively). The trenched structures filled with polysilicon and oxide had a respective breakdown voltage of 790 V (with a space and width of 0.5 μm and 2.0 μm, respectively) and 1320 V (with a space and width of 1.3 μm and 1.2 μm, respectively). The last structure consisting of the FFR and poly-trench had the highest breakdown voltage of 1280 V (with a space and width of 0.7 μm and 1.8 μm, respectively). These results show that the oxide-trench and the structure combining the FFR and the poly-trench improve the breakdown voltage by approximately 29% and 33%, respectively,

compared to the FFR. The proposed trenched structures can support the field profile effectively in a large area and reduce the electric field by all trench rings.

**Author Contributions:** Conceptualization, J.-H.J. and H.-J.L.; data curation, J.-H.J.; formal analysis, J.-H.C., G.-H.K., S.-H.C. and H.-J.L.; investigation, J.-H.J.; project administration, H.-J.L.; writing—original draft preparation, J.-H.J.; writing—review and editing, H.-J.L. All authors have read and agreed to the published version of the manuscript.

**Funding:** This research received no external funding.

**Acknowledgments:** 1. This work was supported by the National Research Foundation of Korea(NRF) grant funded by the Korea government(MSIT) (No. NRF-2017R1A2B2011106). 2. This work was supported by the Comprehensive Technical Support Project for Power Semiconductor Industry Development Project, funded by Busan Metropolitan City and Busan Techno Park, Korea.

**Conflicts of Interest:** The authors declare no conflict of interest.

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
