# Peer review of "Study of a SiC Trench MOSFET Edge-Termination Structure with a Bottom Protection Well for a High Breakdown Voltage"

_applsci, doi:10.3390/app10030753_

Round 1

Reviewer 1 Report

The contents of this paper are useful for researchers working in the field of SiC devices. In this paper the authors have presented a new termination design which involves trenches in the termination region. The paper is well written and results are discussed in details.

The authors, although,  have not described the models which they used  in TCAD simulations. Physics based models for SiC devices are  not well incorporated and developed in these tools as compared to their Si counterparts; so  it is necessary to include the ones which are used for this study. After this minor change the paper will be suitable for publication in this journal.

Author Response

Thank you for the encouraging comments. We upload a response to the reviewer's comment as a Word file. Please see the attchment.

Reviewer 2 Report

The following comments and suggestions are proposed.

Please explain for Figure 3 a.b.c.d why potential distributions are presented with different space and width for different trenched structures and the conventional FFR. How about other space and width for each structure. Please show that it is reasonable and practical to use the parameters in Table 1 for simulation Please explain the variation details of breakdown voltage versus space in Figure 2. There is no verification for the simulation in this manuscript, please provide convincing verification.

Author Response

We are grateful to the revierwer for the insightful comments on my paper. We upload the respnses to the reviewer's comments as a Word file. Please see the attachment.

Reviewer 3 Report

In this paper a novel edge-termination structure for a SiC trench metal–oxide–semiconductor field-effect transistor (MOSFET) power device is proposed. The work presented in this manuscript is useful for the scientific community working in this field. All conclusions are correct and supported by the experiment, analysis and discussion. I recommend to publish this manuscript.
Best regards

Author Response

We appreciate the encouraging comments.

Reviewer 4 Report

The manuscript presents the modeling of MOSFET edge-termination designs to improve the breakdown voltage and how the potential distribution is in these structures. The modeling looks appropriate and worth experimental tests in the future. I agree for publishing this paper. 

Author Response

We appreciate the encouraging comments.

Round 2

Reviewer 2 Report

This manuscript could be considered for acceptance.